# Parents' smoking onset before conception as related to body mass index and fat mass in adult offspring: Findings from the RHINESSA generation study

Gerd Toril Mørkve Knudsen[1,2]*, Shyamali Dharmage[3], Christer Janson[4], Michael J. Abramson[5], Bryndís Benediktsdóttir[6,7], Andrei Malinovschi[8], Svein Magne Skulstad[2], Randi Jacobsen Bertelsen[1,9], Francisco Gomez Real[1], Vivi Schlünssen[10,11], Nils Oskar Jõgi[1,2,12], José Luis Sánchez-Ramos[13], Mathias Holm[14], Judith Garcia-Aymerich[15,16,17], Bertil Forsberg[18], Cecilie Svanes[2,19‡], Ane Johannessen[19‡]

1 Department of Clinical Science, University of Bergen, Bergen, Norway, 2 Department of Occupational Medicine, Haukeland University Hospital, Bergen, Norway, 3 School of Population and Global Health, The University of Melbourne, Carlton, Australia, 4 Department of Medical Sciences: Respiratory, Allergy and Sleep Research, Uppsala University, Uppsala, Sweden, 5 School of Public Health & Preventive Medicine, Monash University, Melbourne, Australia, 6 Faculty of Medicine, University of Iceland, Reykjavik, Iceland, 7 Department of Sleep Medicine, Landspitali, Reykjavik, Iceland, 8 Department of Medical Sciences: Clinical Physiology, Uppsala University, Uppsala, Sweden, 9 Oral Health Center of Expertise in Western Norway, Hordaland, Bergen, Norway, 10 Department of Public Health, Work, Environment and Health, Danish Ramazzini Centre, Aarhus University Denmark, Aarhus, Denmark, 11 National Research Centre for the Working Environment, Copenhagen, Denmark, 12 Lung Clinic, Tartu University Hospital, Tartu, Estonia, 13 Department of Nursing, Huelva University, Huelva, Spain, 14 Occupational and Environmental Medicine, School of Public Health and Community Medicine, Institute of Medicine, University of Gothenburg, Gothenburg, Sweden, 15 ISGlobal, Barcelona, Spain, 16 Universitat Pompeu Fabra (UPF), Barcelona, Spain, 17 CIBER Epidemiología y Salud Pública (CIBERESP), Madrid, Spain, 18 Department of Public Health and Clinical Medicine, Umeå University, Umeå, Sweden, 19 Centre for International Health, Department of Global Public Health and Primary Care, University of Bergen, Bergen, Norway

‡ These authors are joint senior authors on this work.
* gerd.knudsen@uib.no

**Data Availability Statement:** Due to Norwegian ethical and legal restrictions the data underlying

## Abstract

Emerging evidence suggests that parents' preconception exposures may influence offspring health. We aimed to investigate maternal and paternal smoking onset in specific time windows in relation to offspring body mass index (BMI) and fat mass index (FMI). We investigated fathers (n = 2111) and mothers (n = 2569) aged 39–65 years, of the population based RHINE and ECRHS studies, and their offspring aged 18–49 years (n = 6487, mean age 29.6 years) who participated in the RHINESSA study. BMI was calculated from self-reported height and weight, and FMI was estimated from bioelectrical impedance measures in a subsample. Associations with parental smoking were analysed with generalized linear regression adjusting for parental education and clustering by study centre and family. Interactions between offspring sex were analysed, as was mediation by parental pack years, parental BMI, offspring smoking and offspring birthweight. Fathers' smoking onset before conception of the offspring (onset $\geq$15 years) was associated with higher BMI in the offspring when adult (β 0.551, 95%CI: 0.174–0.929, p = 0.004). Mothers' preconception and postnatal smoking onset was associated with higher offspring BMI (onset <15 years:

this study are available upon request to qualified researchers. Requests for data access can be directed to Haukeland University Hospital, 5021 Bergen, Norway. Att. Head of Department. Dept. of Occupational Medicine, Marit Grønning; email: postmottak@helse-bergen.no; phone: +47 55975000. Org, nr. 983 974 724.

**Funding:** Co-ordination of the RHINESSA study has received funding from the Research Council of Norway (Grants No. 274767, 214123, 228174, 230827 and 273838), ERC StG project BRuSH #804199, the European Union's Horizon 2020 research and innovation program under grant agreement No. 633212 (the ALEC Study WP2), the Bergen Medical Research Foundation, and the Western Norwegian Regional Health Authorities (Grants No. 912011, 911892 and 911631). Study centres have further received local funding from the following: Bergen: the above grants for study establishment and co-ordination, and, in addition, World University Network (RDF and Sustainability grant), Norwegian Labour Inspection, and the Norwegian Asthma and Allergy Association. Albacete and Huelva: SEPAR. Fondo de Investigación Sanitaria (FIS PS09). Gøteborg, Umeå and Uppsala: the Swedish Lung Foundation, the Swedish Asthma and Allergy Association. Reykjavik: Iceland University. Melbourne: NHMRC, Melbourne University, Tartu: the Estonian Research Council (Grant No. PUT562). Århus: The Danish Wood Foundation (Grant No. 444508795), the Danish Working Environment Authority (Grant No. 20150067134). The RHINE study received funding by Norwegian Research Council, Norwegian Asthma and Allergy Association, Danish Lung Association, Swedish Heart and Lung Foundation, Vårdal Foundation for Health Care Science and Allergy Research, Swedish Asthma and Allergy Association, Swedish Lung Foundation, Icelandic Research Council, and Estonian Science Foundation. The co-ordination of ECRHS was supported by European Union's Horizon 2020 research and innovation program under grant agreement No. 633212 (the ALEC Study), the European Commission frameworks 5 and 7 (ECRHS I and II) and the Medical Research Council (ECRHS III).

**Competing interests:** The authors have declared that no competing interests exist.

β1.161, 95%CI 0.378–1.944; onset ≥15 years: β0.720, 95%CI 0.293–1.147; onset after offspring birth: β2.257, 95%CI 1.220–3.294). However, mediation analysis indicated that these effects were fully mediated by parents' postnatal pack years, and partially mediated by parents' BMI and offspring smoking. Regarding FMI, sons of smoking fathers also had higher fat mass (onset <15 years β1.604, 95%CI 0.269–2.939; onset ≥15 years β2.590, 95%CI 0.544–4.636; and onset after birth β2.736, 95%CI 0.621–4.851). There was no association between maternal smoking and offspring fat mass. We found that parents' smoking before conception was associated with higher BMI in offspring when they reached adulthood, but that these effects were mediated through parents' pack years, suggesting that cumulative smoking exposure during offspring's childhood may elicit long lasting effects on offspring BMI.

## Background

Maternal smoking during pregnancy plays a significant role in increased risk of obesity and metabolic disorders in the offspring [1–4]. Nicotine and other tobacco constituents cross the placenta, and impair foetal growth [5, 6], which together with determinants such as low birthweight and subsequent rapid postnatal weight gain have been associated with risk of adiposity later in life [4]. Several epidemiological studies also report independent effects of paternal smoking (during pregnancy or postnatal life) associated with greater offspring BMI, body fat distribution and increased risk of overweight in children [7–12]. However, obesity is a complex multifactorial condition with a wide range of determinants, which besides environmental factors, also include behavioural and genetic components.

Recent evidence suggests that the germline cells of the parents might have critical exposure-sensitive periods for triggering epigenetic responses that can affect subsequent offspring's metabolic health and risk of becoming obese [13–15], thus suggesting an epigenetic basis of variation in BMI levels and fat mass. Observations from the Överkalix and ALSPAC cohorts showed that excess food supply and smoking during mid-childhood and pre-pubertal years were associated with metabolic and cardiovascular health, and risk of becoming obese in subsequent generation(s) [16–19]. These findings remain to be successfully replicated, and there exists a possibility of residual confounding due to unmeasured family factors, especially due to the social patterning and inequalities related to smoking behaviour [20, 21]. However, other epidemiological studies have reported adverse offspring outcomes related to paternal exposures in pre-puberty/puberty. Analyses of the RHINESSA, RHINE and ECRHS cohorts found that asthma was more common in offspring with fathers who were obese in puberty [22], as well as in offspring with fathers who smoked in adolescent years [23, 24].

With regard to sex-specific patterns, some studies report no sex differences in offspring BMI in relation to parental smoking [9, 25–27]. Other epidemiological [7, 8, 28] and experimental studies [29–32] indicate more pronounced effects among female offspring. In contrast, the ALSPAC study, reported associations between paternal smoking and increased risk of obesity to be significant only in the sons [16, 19]. Whether sexual dimorphism may be involved in parental transmission of smoking effects on offspring BMI, thus needs further investigation.

The aims of the present study were firstly, to investigate parental smoking onset in specific time windows (onset before 15 years; from age 15 and before conception; after offspring birth) in relation to offspring BMI and, in a subsample, fat mass. Secondly, we aimed to explore whether effects of preconception and early life parental smoking on offspring overweight was

modified by sex of the offspring, and mediated by parental pack years of smoking, parental BMI, offspring smoking and, in a subsample, offspring birthweight.

## Methods

### Study design and population

We investigated onset of parental smoking in relation to adult offspring BMI, using information from two generations. Data concerning the parent population were obtained from the population-based studies Respiratory Health in Northern Europe study (RHINE, www.rhine.nu) and the European Community Respiratory Health Survey (ECRHS, www.ecrhs.org). Information regarding their offspring were collected in the RHINESSA study (www.rhinessa.net). Medical research committees in each study centre approved the study protocols according to national legislation, and each participant gave written informed consent prior to participation (S1 File).

### Parent population

The parent sample comprised subjects originating from the ECRHS postal survey in 1990–94. The participants from seven Northern European study centres (Reykjavik in Iceland, Bergen in Norway, Umea, Uppsala and Gothenburg in Sweden, Aarhus in Denmark, and Tartu in Estonia) were followed up in the RHINE questionnaire study, 10 and 20 years after this baseline survey. At each study wave, postal questionnaire information was collected on lifestyle habits, sociocultural factors, and environmental factors such as childhood and adult exposure to tobacco smoke. A sub-sample was invited for clinical investigation and interview in the ECRHS follow-up studies after 10 and 20 years. For parents in two Spanish centres (Albacete and Huelva) and one Australian centre (Melbourne), information from ECRHS was harmonized with the RHINE data. The questionnaire forms used in ECRHS and RHINE can be found at http://www.ecrhs.org/Quests/ECRHSIImainquestionnaire.pdf and http://rhine.nu/pdf/rhine%20Norway.pdf/ http://rhine.nu/pdf/ECRHS%20II%20Norway.pdf.

A flowchart of the study population is provided in Fig 1.

### Offspring population

The RHINESSA study (www.rhinessa.net) includes adult offspring (> 18 years) of parents from seven RHINE study centres in Denmark, Iceland, Norway, Sweden and Estonia, and two Spanish (Huelva and Albacete) and one Australian (Melbourne) ECRHS centres. The offspring answered web-based and/or postal questionnaires in 2013–2015, which were harmonized with the RHINE protocols. Sub-samples of offspring who had parents with available clinical information, were invited for clinical investigation and interview, following standardized protocols harmonized with the ECRHS protocols. The questionnaire form used in the RHINESSA can be found at https://helse-bergen.no/seksjon/RHINESSA/Documents/RHINESSA%20Screening%20questionnaires%20adult%20offspring.pdf.

### Exposure: Parental smoking

Parental smoking onset was defined from the questions: *i. "Are you a smoker?" ii. "Are you an ex-smoker?" iii. "If yes "How old were you when you started smoking?" iiii. "Smoked for . . . years." iv. "Stopped smoking in [year]"*. Ever-smokers were categorised according to age at smoking initiation (<15 years/≥15 years), and whether smoking started before conception (≥2 years before offspring birth year) or after the offspring was born (≥1 year after offspring birth year). Thus, we constructed a four-level exposure variable with the mutually exclusive

Parent information is from questionnaires. Offspring information obtained from both questionnaires and clinical examinations.

**Fig 1. Flow chart of study population.** Overview of eligible unique RHINE/ECRHS parents and their RHINESSA offspring, and number excluded due to missing information on offspring's BMI and parental smoking habits.

categories: never smoked, started smoking before age 15 years, started smoking between age 15 years and conception (preconception), and started smoking after offspring birth (postnatal). Parent-offspring pairs for which parents started smoking during the two-year interval around pregnancy and conception (up to 15 months before conception and up to 1 year after birth of the child) were excluded from the analysis (n = 92).

## Outcomes: Offspring body mass index and fat mass index

Body mass index (BMI) was calculated from self-reported height and weight [weight (kg)/ height (m)$^2$]. Body composition and fat mass were estimated from bioelectrical impedance analysis measured using Bodystat 1500 MDD (https://www.bodystat.com/medical/). Fat mass index (FMI) was calculated as fat mass (kg)/height (m)$^2$.

## Potential confounders and mediators

Parental/offspring education was used as a proxy for socioeconomic status and categorised as lower (primary school), intermediate (secondary school) or higher (college or university). Parental pack years pre-conception/ from birth until age 18 years were calculated by multiplying the number of 20-packs of cigarettes smoked per day by the number of years the person had smoked up to ≥2 years before offspring birth year/ up to the offspring's eighteenth birth year. Parental BMI was calculated from self-reported height and weight at RHINE III. Offspring smoking was defined as ever smoking (current/ex-smokers) or never smoking based on the questions *i. "Do you smoke?"* ii. *"Did you smoke previously?"*. Offspring birthweight were obtained from national registry data for a subsample of 813 mother-offspring pairs.

## Statistical analysis

Maternal and paternal lines were analysed separately. Generalized linear regressions were used to analyse the associations between parental smoking in specific time windows and offspring BMI (and FMI in a subsample of 240), with adjustment for parental education. Two-dimensional clustering accounted for study centre and family. We set the Heteroscedasticity Consistent Covariance Matrix (HCCM), to version HC1, which made a degree of freedom correction that inflated each residual by the factor $\sqrt{N/(N-K)}$.

We tested for interactions between offspring sex and parental smoking onset on offspring BMI; the significance level for interaction effects was set to 0.05. We generated regression models and table/figure outputs by use of the 'jtool' package [33]. We considered other covariates, such as parental age, offspring education, the other parent's smoking habits, and BMI (data on the parent who did not participate in RHINE/ECRHS were obtained from the offspring themselves), to be included in the statistical model, as shown in S1 Fig. However, we did not find these factors likely to confound the relationship between parental smoking onset and offspring BMI, and therefore did not include them in the final models.

We constructed mediation models [34, 35] to investigate whether significant associations between parental smoking onset and offspring BMI were influenced by the following mediators: i. parental pack years, ii. parental BMI, iii. offspring smoking (never-smoked / ever smoked), and iv. offspring birthweight (only available for a subsample of offspring). To investigate whether effects differed by gender, we tested for effect modification by offspring sex. We conducted mediation analysis with the R package "Medflex"[36], embedded within the counterfactual framework, as this provided means to infer and interpret direct and indirect effect estimates in a nonlinear setting. Thus, the total effect of an exposure was decomposed into a natural direct effect (the part of the exposure effect not mediated by a given set of potential

mediators) and natural indirect effect (the part of the exposure effect mediated by a given set of potential mediators). We followed the imputation-based approach for expanding and imputing the data and fitted a working model for the outcome mean. We fitted separate natural effect models, specified with robust standard errors based on the sandwich estimator. We generated confidence interval plots to visualise the effect estimates and their uncertainty.

We performed all analyses using R version 3.5.2, downloaded at the Comprehensive R Archive Network (CRAN) at http://www.R-project.org/.

## Results

Of unique fathers, 10% started smoking before age 15 years, 40% started smoking from age 15 years, and 2% started smoking after offspring birth. In the maternal line, 11% started smoking <15 years, 39% started smoking ≥15 years, and 3% started smoking after offspring birth. Fathers and mothers who started smoking prior to conception had higher current BMI and less education compared to never smoking parents (S1A and S1B Table). In both the paternal (n = 2111) and maternal (n = 2569) lines, daughters had higher education, lower current BMI, and higher FMI, and started smoking earlier compared to sons (Table 1A and 1B). In the maternal line, daughters had lower birthweight. Offspring of smoking parents had higher BMI, more frequently smoked themselves and had smoked more years, compared to offspring of never smoking parents. Sons with fathers who started smoking from age 15 but before conception also had higher FMI than sons with never smoking fathers.

### Fathers' smoking onset and offspring BMI and FMI

In unadjusted analyses, father's preconception smoking, both starting before or from age 15 years, was associated with increased offspring BMI (Fig 2). There was no significant interaction between offspring sex and fathers' smoking onset with regard to offspring BMI (p = 0.395). With adjustment for father's education and offspring sex, father's smoking onset ≥15 years was significantly associated with increased BMI in their adult offspring (Table 2 and Fig 2). However, there was no association between postnatal smoking onset and offspring BMI.

In the subsample with data on FMI, father's preconception and postnatal smoking onset were associated with increased offspring FMI (Table 3 and Fig 3). There were significant differences between sons and daughters, and only sons of fathers' who started to smoke ≥15 years of age (interaction p = 0.014) or after birth (interaction p = 0.020) had significantly higher FMI compared to sons of never smoking fathers. This trend was not seen among daughters, however, analysis indicated that both sons and daughters of fathers who started to smoke before the age of 15 had higher fat mass (Table 3 and Figs 3 and 4).

### Mothers' smoking onset and offspring BMI and FMI

Mother's smoking starting at all time points were associated with increased BMI in her offspring (Table 4 and Fig 5). There were no significant differences between sons and daughters, except that sons of mothers who started to smoke ≥15 years (interaction p = 0.010) had significantly higher BMI compared to sons of never smoking mothers. There was no such trend among daughters. There was no association with mothers' preconception and postnatal smoking onset and FMI in her offspring (S2 Table).

### Mediation analyses of fathers' smoking onset and offspring BMI

For the association of father's smoking onset ≥15 years with offspring BMI, we analysed mediation by fathers' pack years of smoking, fathers' BMI, and offspring's smoking (Table 5

**Table 1.** A. Characteristics of 2111 fathers with 2939 sons and daughters. B. Characteristics of 2569 mothers with 3548 sons and daughters.

A

|  | Sons | Daughters | P-value |
|---|---|---|---|
|  | N = 1255 (43) | N = 1684 (57) |  |
| **Paternal characteristics** |  |  |  |
| Age years, mean ± SD | 55.1 ± 6.2 | 55.0 ± 6.0 | p = 0.26 |
| Range | 39–65 | 39–65 |  |
| BMI kg/m$^2$, mean ± SD | 26.9 ± 3.8 | 26.8 ± 3.7 | p = 0.32 |
| Range | 16.5–53.3 | 16.8–53.7 |  |
| Educational level, n (%) |  |  |  |
| Primary | 186 (15) | 267 (16) | p = 0.70 |
| Secondary | 466 (37) | 617 (37) |  |
| University/College | 588 (47) | 792 (47) |  |
| Smoking status, n (%) |  |  |  |
| Never smoked | 616 (49) | 783 (47) | p = 0.20 |
| Preconception <15smoking onset | 126 (10) | 179 (11) |  |
| Preconception ≥15 smoking onset | 482 (38) | 696 (41) |  |
| Postconception smoking onset | 31 (3) | 26 (2) |  |
| Years smoked, mean ± SD | 12.0 ± 15.4 | 12.4 ± 15.0 | p = 0.33 |
| Range | 0–59 | 0–52 |  |
| Packyears up to offspring age 18, median | 17.4 | 16.7 | p = 0.95 |
| 25$^{th}$%, 75$^{th}$% | 8.0, 27.2 | 9.9, 25.0 |  |
| Packyears preconception years, median | 7.0 | 7.0 | p = 0.95 |
| 25$^{th}$%, 75$^{th}$% | 3.8, 12.0 | 4.0, 11.7 |  |
| Age smoking onset, mean ± SD | 17.6 ± 5.5 | 17.0 ± 4.5 | p = 0.10 |
| Range | 6–53 | 7–50 |  |
| **Offspring characteristics** |  |  |  |
| Age years, mean ± SD | 29.5 ± 7.4 | 29.7 ± 7.3 | p = 0.53 |
| Range | 18–49 | 18–50 |  |
| BMI kg/m$^2$, mean ± SD | 25.1 ± 4.2 | 23.8 ± 4.8 | p < 0.01 |
| Range | 15.8–52.5 | 14.3–67.2 |  |
| FMI fat mass kg/m$^2$, mean ± SD | 4.7 ± 2.9 | 5.9 ± 2.4 | p < 0.01 |
| Range | 1.1–11.7 | 2.5–14.4 |  |
| Educational level, n (%) |  |  |  |
| Primary | 41 (3) | 40 (2) | p < 0.01 |
| Secondary | 567 (45) | 550 (33) |  |
| University/College | 644 (51) | 1089 (65) |  |
| Smoking status, n (%) |  |  |  |
| Never | 886 (71) | 1174 (70) | p = 0.43 |
| Ever | 363 (29) | 503 (30) |  |
| Years smoked, mean ± SD | 9.4 ± 7.0 | 9.2 ± 7.0 | p = 0.79 |
| Range | 0–36 | 0–33 |  |
| Age smoking onset, mean ± SD | 16.9 ± 2.9 | 16.2 ± 2.7 | p < 0.01 |
| Range | 9–28 | 10–30 |  |

B

|  | Sons | Daughters | p-value |
|---|---|---|---|
|  | N = 1522 (43) | N = 2026 (57) |  |
| **Maternal characteristics** |  |  |  |
| Age years, mean ± SD | 54.3 ± 6.6 | 54.1 ± 6.4 | p = 0.27 |

*(Continued)*

**Table 1.** (Continued)

| | | | |
|---|---|---|---|
| Range | 39–65 | 39–65 | |
| BMI kg/m$^2$, mean ± SD | 25.5 ± 4.3 | 25.7 ± 4.6 | p = 0.19 |
| Range | 14.2–49.3 | 16.8–65.5 | |
| Educational level, n (%) | | | |
| Primary | 197 (13) | 361 (18) | p < 0.01 |
| Secondary | 542 (36) | 659 (33) | |
| University/College | 773 (51) | 999 (49) | |
| Smoking status, n (%) | | | |
| Never smoked | 732 (48) | 965 (48) | p = 0.42 |
| Preconception <15smoking onset | 154 (10) | 232 (12) | |
| Preconception ≥15 smoking onset | 594 (39) | 780 (39) | |
| Postconception smoking onset | 42 (3) | 49 (2) | |
| Years smoked, mean ± SD | 11.1 ± 14.3 | 11.2 ± 14.2 | p = 0.79 |
| Range | 0–52 | 0–41 | |
| Packyears up to offspring age 18, median | 11.5 | 12.5 | p = 0.45 |
| 25$^{th}$%, 75$^{th}$% | 5.8, 18.8 | 6.0, 19.2 | |
| Packyears preconception years, median | 4.2 | 5.0 | p = 0.01 |
| 25$^{th}$%, 75$^{th}$% | 2.5, 7.0 | 3.0, 8.0 | |
| Age smoking onset, mean ± SD | 17.3 ± 4.3 | 17.0 ± 4.0 | p = 0.22 |
| Range | 9–49 | 7–44 | |
| **Offspring characteristics** | | | |
| Age years, mean ± SD | 31.0 ± 7.8 | 30.9 ± 7.7 | p = 0.49 |
| Range | 18–52 | 18–52 | |
| Birthweight kg, mean ± SD | 3.5 ± 0.6 | 3.4 ± 0.6 | p < 0.01 |
| Range | 1.1–5.3 | 0.5–5.3 | |
| BMI kg/m$^2$, mean ± SD | 25.3 ± 3.9 | 23.8 ± 4.4 | p < 0.01 |
| Range | 12.7–44.7 | 14.9–49.0 | |
| FMI fat mass kg/m$^2$, mean ± SD | 4.0 ± 1.7 | 7.3 ± 4.3 | p <0.01 |
| Range | 1.0–6.6 | 3. 0–20.5 | |
| Educational level, n (%) | | | |
| Primary | 45 (3) | 49 (2) | p < 0.01 |
| Secondary | 650 (43) | 651 (32) | |
| University/College | 826 (54) | 1321 (65) | |
| Smoking status, n (%) | | | |
| Never | 1023 (67) | 1321 (65) | p = 0.32 |
| Ever | 493 (32) | 699 (35) | |
| Years smoked, mean ± SD | 9.4 ± 7.1 | 9.7 ± 7.2 | p = 0.49 |
| Range | 0–37 | 0–35 | |
| Age smoking onset, mean ± SD | 16.6 ± 3.1 | 16.0 ± 2.7 | p < 0.01 |
| Range | 7–32 | 10–36 | |

Test for sign differences between offspring sex; Wilcoxon Mann Whitney test for continuous variables, chi square and Kruskal Wallis test for categorical variables.

Missing paternal values: Age: NA = 37; BMI: NA = 34; Educational level: NA = 23; Packyears. NA = 836. Missing offspring values: Age: NA = 7, FMI: NA = 2812, Educational level: NA = 8; Smoking status: NA = 13; Years smoked: NA = 72; Age smoking onset: NA = 29.

Missing maternal values: Age: NA = 80; BMI: NA = 85; Educational level: NA = 17; Packyears: NA 868. Missing offspring values: Age: NA = 10; FMI: NA = 3440, Educational level: NA = 6; Smoking status: NA = 12; Years smoked: NA = 63; Age smoking onset: NA = 25. Birthweight only available in subsample n = 813 (335 males and 478 females)

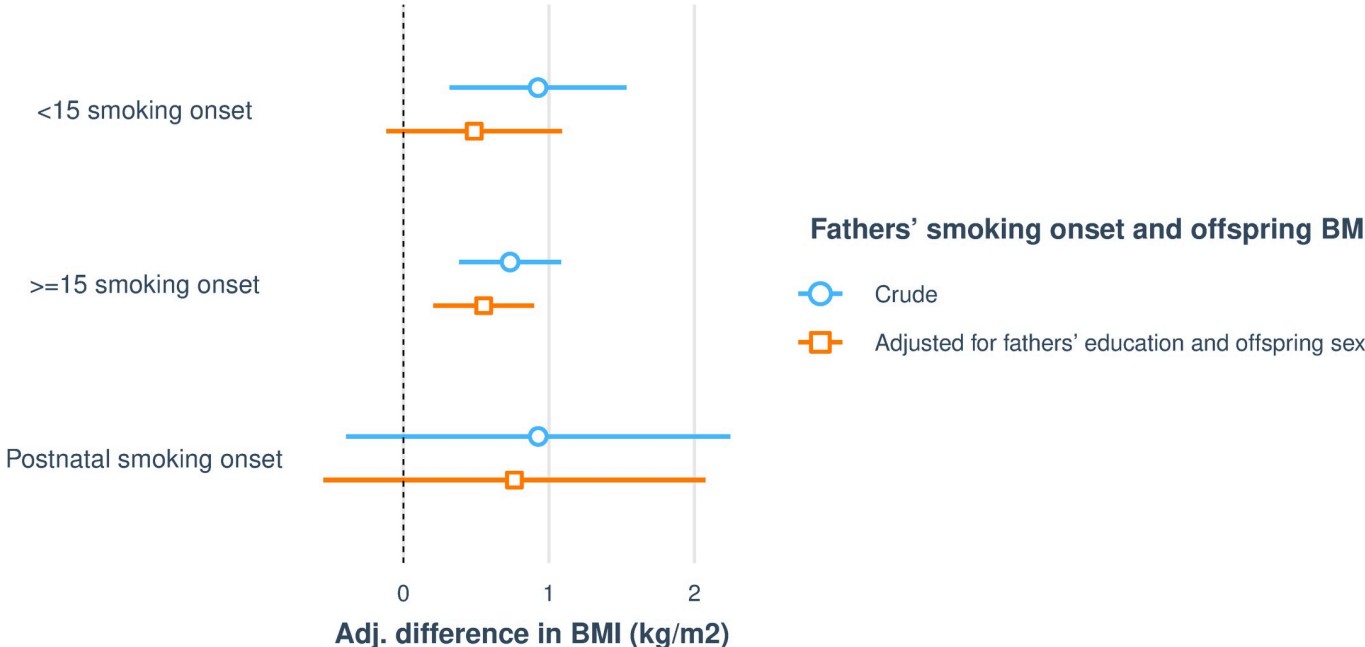

**Fig 2. Visualising associations between fathers' smoking onset and offspring (n = 2916) BMI.** The figure shows crude regressions and regressions adjusted for fathers' education and offspring sex. After adjustment for fathers' education, fathers' smoking onset ≥ 15 remains significantly associated with increased BMI in offspring.

and S2 Fig). Mediation analysis by fathers' pack years up to offspring age 18 revealed indirect but no direct effect, thus suggesting full mediation of the observed association between fathers' smoking onset ≥ 15 years and offspring BMI by fathers' pack years. When restricting analysis to pack years in preconception years only, there was no mediation via fathers' accumulative smoking.

Mediation by fathers' BMI confirmed both a direct effect of fathers' smoking onset ≥15 years and an indirect effect via fathers' BMI, suggesting partial mediation by fathers' BMI.

Similarly, there was partial mediation of the association between fathers' smoking onset ≥ 15 years and offspring obesity by offspring smoking status with both a direct and an indirect effect.

None of the above observed effects were modified by offspring sex.

**Table 2. Associations between fathers' smoking onset and offspring (n = 2916) BMI.**

| | Sons' and daughter's BMI | | |
|---|---|---|---|
| *Predictors (kg/m²)* | *Adj. difference in BMI* | *95% CI* | *P* |
| Preconception smoking onset < 15 years of age n = 303 | 0.486 | -0.196–1.169 | 0.162 |
| Preconception smoking onset ≥ 15 years of age n = 1162 | 0.551 | 0.174–0.929 | 0.004** |
| Postnatal smoking onset n = 57 | 0.763 | -0.692–2.217 | 0.304 |

Estimates from generalized linear regression models with adjustment for offspring sex and fathers' education. Clustered by family id and study centre. P value significance level: *.05,

**.01, ***.001.

When adjusting for fathers' education, fathers' smoking onset ≥15 remains significantly associated with increased BMI in offspring.

**Table 3. Associations between fathers' smoking onset and offspring (n = 129) FMI.**

| | Sons' and daughter's FMI | | | |
|---|---|---|---|---|
| Predictors | Adj. difference in FMI (fat mass kg/m²) | 95% CI | P | Interaction sex P |
| Preconception smoking onset < 15 years of age | 1.604 | 0.269–2.939 | 0.019 ** | 0.982 |
| Preconception smoking onset ≥ 15 years of age[a] | 2.590 | 0.544–4.636 | 0.013 ** | 0.014 ** |
| Postnatal smoking onset[b] | 2.736 | 0.621–4.851 | 0.011 ** | 0.020 ** |

[a]moking onset ≥15: daughters β: -2.797, CI: (-5.023, -0.571)

[b] Postnatal smoking onset: daughters β: -3.041, CI: (-5.599, -0.483)

Estimates from generalized linear regression models with offspring sex as interaction term and adjustment for fathers' education.

Clustered by family id and study centre. P value significance level: *.05,

**.01, ***.001

## Mediation analyses of mothers' smoking onset and offspring BMI

With regard to the maternal line, there were significant associations of mother's smoking starting <15 years, ≥15 years, and postnatally, thus, we analysed mediation by mothers' pack years of smoking, mothers' BMI and offspring smoking for each of these associations.

Similarly to the mediation analyses in the paternal line, mediation analysis by mothers' pack years up to offspring's age 18 revealed presence of an indirect but no direct effect, suggesting full mediation of the observed association between mother's preconception smoking onset both before and from 15 years, and offspring BMI (onset <15 years: β: 1.059, p <0.001; onset ≥15 years β: 0.833, p <0.001; S3 Table and S3 Fig). There was partial mediation of mothers' pack years up to offspring's age 18 on mothers' postnatal smoking onset and offspring BMI where both indirect (β: 0.276, p = 0.001) and direct (β: 1.950, p <0.001) effects were

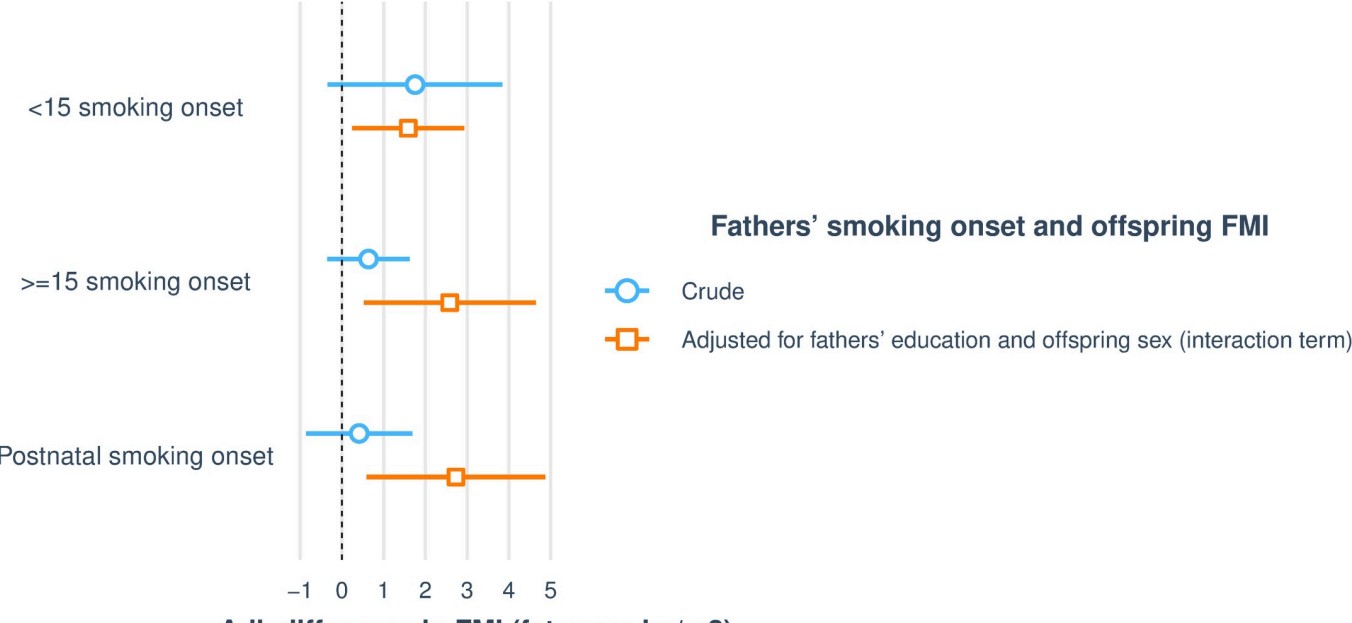

**Fig 3. Visualising associations between fathers' smoking onset and offspring (n = 129) FMI.** The figure shows crude regressions and regressions adjusted for fathers' education and offspring sex added as an interaction term. In fully adjusted model, fathers' smoking onset at all time points (<15, ≥ 15 and after birth) are significantly associated with increased FMI in offspring, but there are significant differences between offspring sex, and only sons of fathers who started to smoke ≥15 years of age (interaction p = 0.014) or after birth (interaction p = 0.020) had significantly higher FMI compared to sons of never smoking fathers.

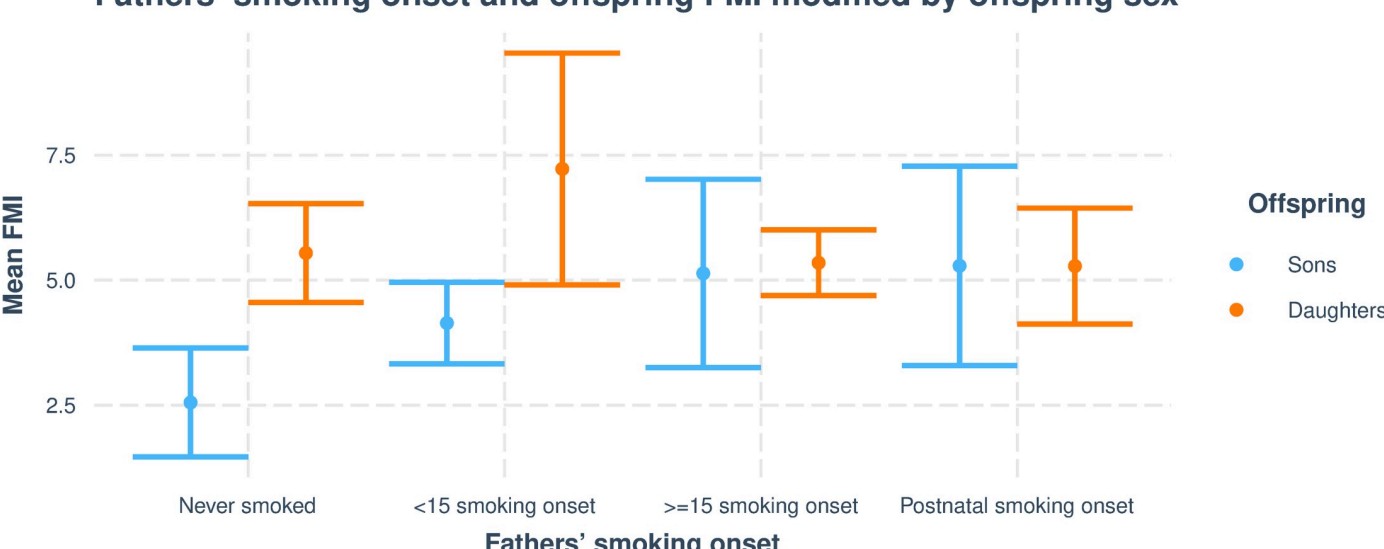

**Fig 4. Visualising mean FMI differences in sons and daughters according to fathers' smoking onset.** Interaction plot, depicting how offspring sex modify the associations between fathers' ≥15 and postnatal smoking onset and offspring's FMI.

significant and pointed in the same direction. We did not find any direct or indirect effects via mothers' preconception pack years (S3 Table and S3 Fig).

Mediation by mothers' BMI confirmed partial mediation with presence of both a direct effect of mothers' preconception smoking onset before 15 years of age (β: 0.551, p = 0.026) as well as smoking onset after birth (β: 1.869, p <0.001), and an indirect effect via mothers' BMI (onset <15: β: 0.334, p <0.001; onset after birth: β: 0.320, p = 0.013). There was no evidence of direct or indirect effects via mothers' BMI in relation to mothers' preconception smoking onset ≥15 (S4 Table and S4 Fig).

There was indication of partial mediation by offspring's own smoking status, as both direct effects of smoking onset before 15 years of age (β: 0.841, p = 0.001) and smoking onset after birth (β: 2.090, p <0.001), as well as indirect effects via offspring's smoking were present (onset <15 years: β: 0.059, p = 0.016; onset ≥15 years β: 0.031, p = 0.019; onset after birth: β: 0.129, p = 0.013, S5 Table and S5 Fig).

**Table 4. Associations between mothers' smoking onset and offspring (n = 3531) BMI.**

| | Sons' and daughter's BMI | | | |
|---|---|---|---|---|
| *Predictors* | *β-coef.* | *95% CI* | *P* | *Interaction sex P* |
| Preconception smoking onset < 15 years of age | 1.161 | 0.378–1.944 | 0.004 ** | 0.338 |
| Preconception smoking onset ≥ 15 years of age[a] | 0.720 | 0.293–1.147 | 0.001 ** | 0.010 ** |
| Postnatal smoking onset | 2.257 | 1.220–3.294 | <0.001 *** | 0.952 |

[a] Smoking onset ≥15: daughters β: -0.717, CI: (-1.264, -0.170)

Estimates from generalized linear regression with offspring sex as interaction term and adjustment for mothers' education. Clustered by family id and study centre. P value significance level: *.05,

**.01,

***.001

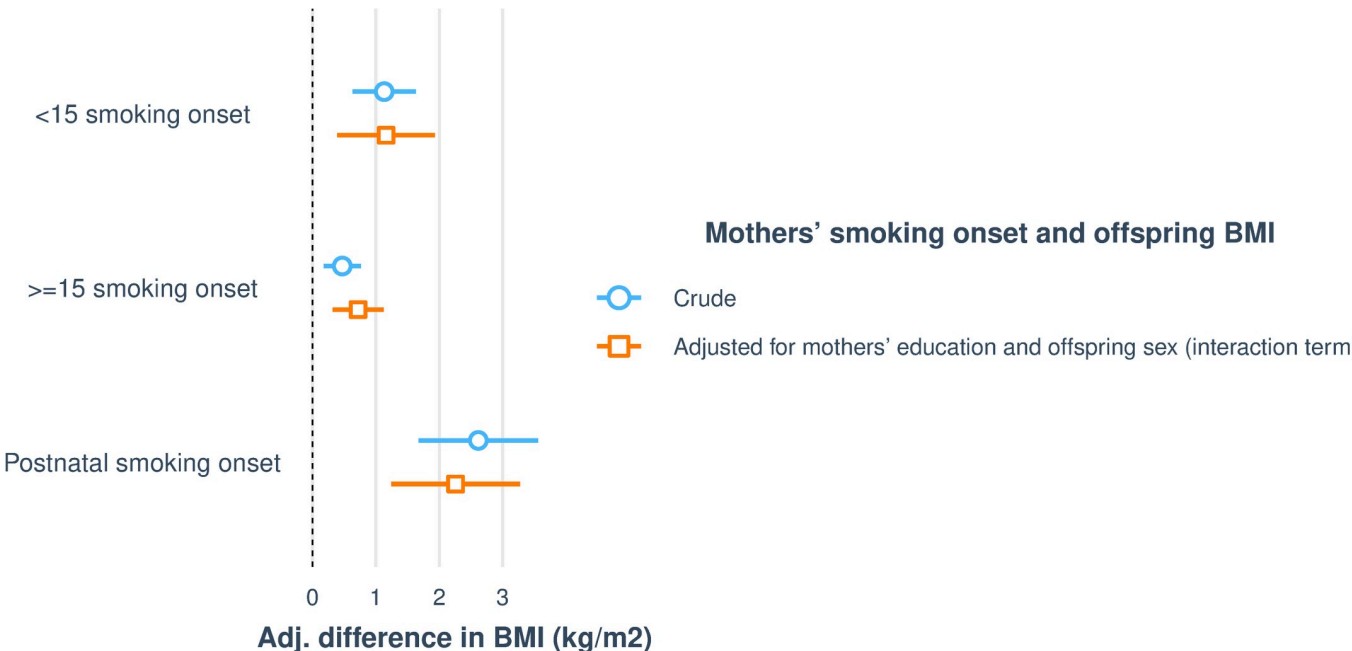

**Fig 5. Visualising associations between mothers' smoking onset and offspring (n = 3531) BMI.** The figure shows crude and adjusted regressions, with adjustment for mothers' education and offspring sex added as interaction term. In fully adjusted model, mothers' smoking onset at all time points (<15, ≥ 15 and after birth) are significantly associated with increased BMI in offspring, but with significant differences between offspring sex. Only sons of mothers who started to smoke ≥15 years (interaction p = 0.010) had significantly higher BMI compared to sons of never smoking mothers.

In a subsample with birth weight data, there was no evidence of mediation by offspring birthweight as only a direct effect of mothers' smoking onset <15 years on offspring BMI were present (S6 Table and S6 Fig).

None of the above observed effects were modified by offspring sex.

## Discussion

Father's smoking starting before conception was associated with higher BMI in his adult offspring. Bioimpedance measurements for a subsample also found that sons of smoking fathers, starting both before conception and during postnatal years, had higher fat mass, thus suggesting a consistent effect on sons' body composition. Mother's preconception and postnatal smoking onset was also associated with higher adult BMI in her offspring, but these associations were not supported by fat mass analysis in a subsample. Mediation analyses showed that the observed associations between parents' preconception smoking onset and offspring BMI were fully mediated via parents' postnatal pack years. Furthermore, parents BMI and offspring's own smoking status partially mediated the effects of parents' smoking onset on offspring BMI.

To our knowledge, this is the first study that has shown consistently higher BMI and fat mass levels in offspring of smoking fathers' where the offspring has reached adulthood. Our results further suggest that fathers' smoking may have more pronounced effects on their sons' fat mass when compared to daughters. A potential sex-specific effect on offspring's body composition supports previous reports of particularly paternal smoking trajectories to impact on sons' fat mass and risk of becoming obese [16, 19]. However, in contrast to findings in the ALSPAC study, where only fathers' smoking in mid-childhood and pre-pubertal years was associated with increased BMI and fat mass in the sons [19], our study indicate that father's

**Table 5. Mediation of the observed association between fathers' ≥15 smoking onset and offspring BMI.**

| Fathers' smoking onset | Causal mediation analysis father offspring | | | |
|---|---|---|---|---|
| | Adj diff. BMI (kg/m$^2$) | Std. error | z value | P value |
| **A) Mediation by fathers' packyears up to offspring age 18** | | | | |
| *Preconception smoking onset ≥15* | | | | |
| Natural direct effect | 0.240 | 0.318 | 0.756 | 0.450 |
| Natural indirect effect | 0.482 | 0.239 | 2.014 | 0.044 * |
| Total effect | 0.722 | 0.237 | 3.047 | 0.002 ** |
| Interaction by offspring sex: 0.209 | | | | |
| **B) Mediation by fathers' preconception packyears** | | | | |
| *Preconception smoking onset ≥15* | | | | |
| Natural direct effect | 0.677 | 0.235 | 2.879 | 0.004 ** |
| Natural indirect effect | - 0.092 | 0.130 | - 0.708 | 0.479 |
| Total effect | 0.585 | 0.205 | 2.848 | 0.004 ** |
| Interaction by offspring sex: 0.913 | | | | |
| **C) Mediation by fathers' BMI** | | | | |
| *Preconception smoking onset ≥15* | | | | |
| Natural direct effect | 0.367 | 0.170 | 2.159 | 0.031 * |
| Natural indirect effect | 0.214 | 0.053 | 4.058 | < 0.001 *** |
| Total effect | 0.582 | 0.178 | 3.264 | 0.001 ** |
| Interaction by offspring sex: 0.528 | | | | |
| **D) Mediation by offspring smoking status** | | | | |
| *Preconception smoking onset ≥15* | | | | |
| Natural direct effect | 0.488 | 0.180 | 2.711 | 0.007 ** |
| Natural indirect effect | 0.080 | 0.028 | 2.900 | 0.004 ** |
| Total effect Interaction by offspring sex: 0.134 | 0.568 | 0.177 | 3.215 | 0.001** |

Effect decomposition on the scale of the linear predictor with standard errors based on the sandwich estimator.
Conditional on fathers' educational level and offspring sex. P value significance level:

*.05,

**.01,

***.001

preconception smoking starting both before and from age 15 years were associated with increased fat mass in his adult sons. This was also seen in sons where fathers started to smoke after birth. This may reflect the direct toxicogenic effects cigarette smoke exert on biological processes involved in metabolic health. Previous studies have found germ cells and elevated reactive oxygen species (ROS) to mediate metabolic phenotypes in offspring [29, 31, 37, 38]. Smoking has also been shown to induce both ROS overproduction as well as epigenetic changes to germ cells [29, 39], which adds biological plausibility of paternal smoking to be drivers of complex offspring phenotypes. Although increased adipose tissue does not necessarily translate into metabolic abnormalities, both BMI and FMI are regarded important determinants of metabolic health at the population level [40, 41], and childhood adiposity has been reported to be associated with increased risk of adult type 2 diabetes mellitus [42]. In a recent epigenome-wide association study, we found that adult offspring with smoking fathers had differential methylation in regions related to innate immune system pathways and fatty acid biosynthesis [43]. These are inflammatory signalling pathways and metabolic signals that have been linked to obesity [44]. However, whether the observed associations between increased BMI and FMI among offspring of smoking fathers relate to metabolic phenotypes needs

further investigation. Our study also indicated that parental smoking exposures transmit through the maternal line, as also mothers' pre- and postnatal smoking onset was related to higher BMI in her adult sons and daughters. However, offspring of smoking mothers did not have a higher fat mass. This may suggest that maternal and paternal smoking trajectories influence their offspring body composition and risk of obesity through different biological mechanisms and pathways.

Through independent mediation analyses, we sought to investigate how parental smoking onset may influence offspring BMI. By including parental pack years as a potential mediator, we aimed to disentangle the effect of parents' smoking onset, and specifically smoking onset before conception, from an accumulative and sustained smoking exposure during peri-and post- natal life. Our findings show that parents' smoking onset influence their offspring BMI via pack years smoked during childhood years, up to the offspring's age 18. This may very well reflect the importance of families' shared environment and the impact lifestyle-related factors, such as dietary habits and physical activity, exert on BMI levels and risk of obesity [45, 46]. This may also explain why both fathers' preconception as well as postnatal smoking onset was associated with increased fat mass in their sons, and why we did not find preconception pack years to mediate the association between parents' smoking onset and offspring BMI.

Furthermore, we found that parents' BMI, partially mediated the effect of pre- and postnatal smoking onset on offspring BMI. Although this may indicate a genetic contribution in body composition, we also found that offspring's smoking status partially mediated the effect of parents smoking onset on their adult BMI, where offspring who had or were smoking themselves, tended to have higher BMI in adulthood compared to offspring who had never smoked. As such, our results may reflect the influence of multiple pathways and the complex interplay between genetics, biology, behaviour, and environment, potentially involved in the aetiology of obesity [47, 48]. These multifactorial aspects may also explain why our results contrast from previous studies related to offspring asthma outcomes in the RHINESSA, RHINE and ECRHS cohorts, where the fathers' pubertal and adolescent years specifically have been shown to constitute an important time window for transmission of paternal lineage exposures [22–24].

Low birthweight due to growth restriction during pregnancy is one factor that has been thought to be on the causal pathway between maternal smoking and offspring's risk of obesity in later life [4]. We found no evidence that the association between mothers' smoking onset and offspring BMI was mediated via her sons' or daughters' birthweight. However, the present study was not able to distinguish true growth retarded newborns from those being born small due to genetic factors, thus a potential causal role of birthweight on overweight in subsequent years warrants further investigation.

A strength of the present study was that the study population originated from two linked inter-generational study cohorts that enables long-term investigation of exposures, across generations and in adult offspring. Further, we used multinational data following standardized protocols. The study also had clear limitations. The main outcome, offspring BMI, was based on self-reported height and weight which can possibly add bias to our estimates. However, we would expect this potential bias to be non-differential, since offspring of smoking and never smoking parents assessed their height and weight in the same manner. There is no reason to believe that offspring of smoking parents would report height and weight any differently than offspring of non-smoking parents. Moreover, studies assessing the validity of self-reported measurements of anthropometric characteristics, have showed that the correlation between self-reported and technician-measured BMI is high (0.92) [49]. Although BMI does not distinguish between lean and fat mass, it is commonly used to determine overweight in clinical research settings as it is closely related with body fat [50, 51]. In addition, we verified our findings in a sub-sample of sons and daughters with clinical data on fat mass. However, this sub-

sample was of limited size, and we did not have sufficient statistical power to conduct mediation analyses of the observed associations between fathers' smoking onset and offspring fat mass. With regard to smoking exposure, we had information only on the participating parent, and have thus not been able to account for a potential smoking exposure arising from the other parent in the household. Neither do we have detailed information about where the parents smoked (inside house/outside house/other places), thus we have not been able to address levels of cigarette smoke the offspring would have been exposed to. Furthermore, we excluded parent-offspring pairs with missing information on parental smoking (n = 1477), which consequently reduced our sample size. Some of the parental smoking onset categories were also limited in numbers, which potentially could influence the reliability of our results. A multitude of exposures and difference in genetic background exists in population studies, and as the offspring in the present study have reached adulthood, they have been exposed to a variety of environmental factors. However, to be regarded as potential confounders, they would per definition precede both the exposure (parental smoking onset in adolescent and early adult years) and outcome (adult offspring BMI) in time. Thus, this does rule out many factors that traditionally would be included in models assessing associations with BMI in adults. We investigated whether parents' adult BMI mediated the effect of parental smoking onset on offspring BMI. However, we did not have information on parents BMI in childhood and pre-adolescent years, which potentially can be of importance and a potential confounder as this would precede both the exposure and outcome in time. Moreover, we did not have information regarding adoption in the offspring population, and whether the participating parent was the biological parent. Thus, unmeasured factors may have impacted on our findings. We chose to use a mediation analysis embedded within the counterfactual framework due to its flexibility in handling non- linear parametric models. However, we have not been able to assess the robustness of our findings and investigated whether there are violations to the identification assumptions, especially with regard to all potential variables being independent and accounted for. This should be further investigated.

## Conclusion

In this multicentre population-based study of two generations, we found that fathers beginning to smoke before conception was associated with higher BMI in their adult sons and daughters, and that father's smoking starting in any time window was associated with higher FMI in adult sons. In contrast, mothers' pre-as well as postnatal smoking onset was associated with higher offspring adult BMI, but not higher fat mass. Independent mediation analysis indicated that parents' pack years up to offspring's age 18, but not preconception pack years, fully mediated these effects. This may suggest that an accumulative smoking exposure during offspring's childhood may be needed in order to elicit long lasting effects on offspring BMI and risk of becoming obese. In addition, we found partial mediation by parents' BMI and offspring own smoking status, which may further reflect the importance of families' shared environment and the impact lifestyle-related factors, such as dietary habits and physical activity, exert on BMI levels and risk of obesity. As such, our results support the multifactorial aspects contributing to obesity.

## Supporting information

**S1 Fig. Directed Acyclic Graph (DAG).** The figure presents covariates considered to be included in the statistical model.
(TIF)

**S2 Fig. Visualising mediations of the association with fathers' ≥15 smoking onset on offspring BMI.** Analyses reveal full mediation by fathers' pack years and partial mediation by fathers' BMI and offspring's own smoking status. There is no mediation via fathers' preconception accumulative smoking.
(TIF)

**S3 Fig. Visualising mothers' pack years as mediator of the observed associations between mothers' smoking onset and offspring BMI.**
(TIF)

**S4 Fig. Visualising mothers' BMI as mediator of the observed associations between mothers' smoking onset and offspring BMI.**
(TIF)

**S5 Fig. Visualising offspring's smoking habits as mediator of the observed associations between mothers' smoking onset and offspring BMI.**
(TIF)

**S6 Fig. Visualising offspring's birthweight as mediator of the observed associations between mothers' smoking onset and offspring BMI.**
(TIF)

**S1 Table.** A. Descriptive table of father offspring cohort grouped by fathers' smoking onset and stratified by offspring sex. B. Descriptive table of mother offspring cohort grouped by mothers' smoking onset and stratified by offspring sex. Parents who started smoking prior to conception have higher current BMI and less education compared to never smoking parents. Offspring of smoking parents have higher BMI, more frequently smoke themselves and have smoked more years, compared to offspring of never smoking parents. Sons with fathers who started smoking from age 15 but before conception also have higher FMI than sons with never smoking fathers.
(PDF)

**S2 Table. Associations between mothers' smoking onset and offspring (n = 111) FMI.** The figure shows regression model adjusted for mothers' education and offspring sex and reveals no association with mothers' preconception and postnatal smoking onset and FMI in her offspring.
(PDF)

**S3 Table. Mothers' pack years as mediator of the observed associations between mothers' smoking onset and offspring BMI.** The association between mothers' preconception smoking onset and offspring BMI is fully mediated by mothers' postnatal pack years, whereas mothers' postnatal smoking onset and offspring BMI is partially mediated by mothers' postnatal pack-years. There is no evidence of direct or indirect effects via mothers' preconception accumulative smoking in relation to mothers' smoking onset.
(PDF)

**S4 Table. Mothers' BMI as mediator of the observed associations between mothers' smoking onset and offspring BMI.** The association between mothers' preconception smoking onset before 15 years of age as well as smoking onset after birth and offspring BMI is partially mediated by mothers' BMI. There is no evidence of direct or indirect effects via mothers' BMI in relation to mothers' preconception smoking onset ≥15.
(PDF)

**S5 Table. Offspring's smoking habits as mediator of the observed associations between mothers' smoking onset and offspring BMI.** The association between mothers' preconception smoking onset before 15 years of age as well as smoking onset after birth and offspring BMI is partially mediated by offspring's own smoking status.
(PDF)

**S6 Table. Offspring's birthweight as mediator of the observed associations between mothers' smoking onset and offspring BMI.** In a subsample with birth weight data, there is no evidence of mediation by offspring birthweight.
(PDF)

**S1 File. Table of ethic committee name and approval number for each study center.**
(PDF)

## Author Contributions

**Conceptualization:** Gerd Toril Mørkve Knudsen, Cecilie Svanes, Ane Johannessen.

**Data curation:** Shyamali Dharmage, Christer Janson, Michael J. Abramson, Bryndís Benediktsdóttir, Andrei Malinovschi, Randi Jacobsen Bertelsen, Francisco Gomez Real, Vivi Schlünssen, Nils Oskar Jõgi, José Luis Sánchez-Ramos, Mathias Holm, Judith Garcia-Aymerich, Bertil Forsberg, Cecilie Svanes, Ane Johannessen.

**Formal analysis:** Gerd Toril Mørkve Knudsen, Cecilie Svanes, Ane Johannessen.

**Funding acquisition:** Shyamali Dharmage, Christer Janson, Bryndís Benediktsdóttir, Randi Jacobsen Bertelsen, Francisco Gomez Real, Vivi Schlünssen, José Luis Sánchez-Ramos, Cecilie Svanes.

**Investigation:** Gerd Toril Mørkve Knudsen, Shyamali Dharmage.

**Methodology:** Gerd Toril Mørkve Knudsen, Shyamali Dharmage, Judith Garcia-Aymerich, Cecilie Svanes, Ane Johannessen.

**Project administration:** Cecilie Svanes, Ane Johannessen.

**Supervision:** Shyamali Dharmage, Svein Magne Skulstad, Cecilie Svanes, Ane Johannessen.

**Visualization:** Gerd Toril Mørkve Knudsen.

**Writing – original draft:** Gerd Toril Mørkve Knudsen, Cecilie Svanes, Ane Johannessen.

**Writing – review & editing:** Gerd Toril Mørkve Knudsen, Shyamali Dharmage, Christer Janson, Michael J. Abramson, Bryndís Benediktsdóttir, Andrei Malinovschi, Svein Magne Skulstad, Randi Jacobsen Bertelsen, Francisco Gomez Real, Vivi Schlünssen, Nils Oskar Jõgi, José Luis Sánchez-Ramos, Mathias Holm, Judith Garcia-Aymerich, Bertil Forsberg, Cecilie Svanes, Ane Johannessen.

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
