## [Decision Letter · Decision Letter 0]

11 May 2020

PONE-D-20-07665

Parents’ smoking onset before conception as related to body mass index and fat mass in adult offspring: Findings from the RHINESSA generation study

PLOS ONE

Dear Mrs Knudsen,

Thank you for submitting your manuscript to PLOS ONE. After careful consideration, we feel that it has merit but does not fully meet PLOS ONE’s publication criteria as it currently stands. Therefore, we invite you to submit a revised version of the manuscript that addresses the points raised during the review process.

Please revise your manuscript according to the reviewers comments.

We would appreciate receiving your revised manuscript by Jun 25 2020 11:59PM. To enhance the reproducibility of your results, we recommend that if applicable you deposit your laboratory protocols in protocols.io, where a protocol can be assigned its own identifier (DOI) such that it can be cited independently in the future. For instructions see: http://journals.plos.org/plosone/s/submission-guidelines#loc-laboratory-protocols

We look forward to receiving your revised manuscript.

Kind regards,

Seana Gall

Academic Editor

PLOS ONE

4. We note you have included a table to which you do not refer in the text of your manuscript. Please ensure that you refer to Table 1 in your text; if accepted, production will need this reference to link the reader to the Table.

Reviewers' comments:

Reviewer's Responses to Questions

**Comments to the Author**

1. Is the manuscript technically sound, and do the data support the conclusions?

Reviewer #1: Partly

Reviewer #2: Yes

2. Has the statistical analysis been performed appropriately and rigorously? 

Reviewer #1: Yes

Reviewer #2: Yes

3. Have the authors made all data underlying the findings in their manuscript fully available?

Reviewer #1: No

Reviewer #2: Yes

4. Is the manuscript presented in an intelligible fashion and written in standard English?

Reviewer #1: Yes

Reviewer #2: Yes

5. Review Comments to the Author

Reviewer #1: Ref: PONE-D-20-07665

Comments to Authors:

Congratulations on this interesting manuscript.

The comments below constitute my review, and a number of points will require addressing.

By way of overall summary: the major revision I suggest is the acknowledgement of the myriad cardiovascular and endocrine risk factors that are all confounders to the outcomes reported here, and as such must be acknowledged as not adjusted for in this work.

Comprehensive review:

In the manuscript titled ‘Parents’ smoking onset before conception as related to body mass index and fat mass in adult offspring: Findings from the RHINESSA generation study’ Knudsen et al report interesting findings related to the largely pre-conception exposure to tobacco smoking in mothers and fathers and subsequent health consequences in offspring through to adulthood. This study comes from analysing three studies, the RHINE and the ECRHS, which provided data on the parents of the cohort, and the RHINESSA project, which provided data on the offspring of this cohort. The RHINESSA study typically publishes data related to respiratory conditions such as asthma and atopic disease. This cohort includes participants from across multiple centres in Europe and also Melbourne, Australia, and aims to follow up multiple generations of participants to investigate genetics and epigenetic influences on disease – as has occurred in this study.

The number of authors listed for such a manuscript is appropriate.

The financial disclosures appear to be thorough and cover the project included in the manuscript.

The ethics statement is complete and appears to cover all research sites.

The authors have responded to the data availability question with ‘no – some restrictions will apply’.

Detailed review:

Background:

The main point of this background is the suggestion that there is evidence that the germline cells of the parents might have critical exposure-sensitive periods for triggering epigenetic responses that will influence the risk of offspring becoming obese later in life. The remainder of the background is not directly of relevance to the manuscript but is interesting. It could be better placed in a broader discussion section as opposed to the background.

The aims of the study are well presented, and ambitious. The overall aim is to investigate how timing of parental smoking onset (divided into 3 groups of before age 15 years, after age 15 years and after birth) may impact offspring BMI and in a small sample, impact fat mass. Secondary aims are to investigate the sex differences in this findings and to investigate the confounders of parental pack years, parental BMI and offspring smoking.

Methods:

Parents: ECRHS survey for baseline data then followed up in RHINE study 10-20 years after. All had postal survey, some had clinics.

Figure 1 is necessary and useful, but can be improved with more details provided about study centres and dates of the original surveys. Currently all we learn from the figure is sample numbers, when it is easy to make this figure more appealing to include the locations of these participants and the dates of the original studies and the form of the studies (postal questionnaires, clinics etc).

Offspring: Appropriately described

Exposure: 4 level exposure created. Parents who started smoking in the 2 year interval around pregnancy were excluded, as the exposure seems only to be defined by years, and not by dates exactly. This is likely due to the original surveys only allowing answers by year.

Outcomes: BMI from self-report. Less than ideal. Body fat mass from bioelectrical impedance analysis and FMI from that measurement/height^2. I am unsure how accurate such techniques are and as such, how valid this approach is.

Confounders: The variables considered were parental socioeconomic status (via parental education), parental pack years, parental BMI (self report), offspring smoking (ever vs never only) and offspring birthweight.

This is a small number of confounders. These are all indeed vital confounders to consider, but there are myriad other critical confounders that could be influencing the outcomes from across the huge list of cardiovascular and endocrine risk factors both prior to conception and post-natally. We also know that the amount of smoking in close confinement to the offspring post birth is of importance, as studies with serum cotinine samples of tobacco exposure prove that although some parents may be smoking a great deal (as per pack years), if they do so outside or otherwise away from the child, they can mitigate possible exposures to their offspring.

In reality a number of other confounders not listed in the methods were also included in results: parental age, offspring education, other parents smoking habits, and other parents BMI (other parents information obtained from the offspring…), however these were not include din final models.

Statistical analysis:

Simple statistical approaches were employed appropriately.

The mediation models were created for all the key confounders listed in the methods. This was performed in an appropriate way utilising well known R packages, which I personally have not utilised.

Results:

Table 1A and 1B and S1A and B: significant missing values for age smoking onset and packyears noted

Tables would be clearer with stepped rows under umbrella terms. For example: ‘educational level, n (%)’ should be far left of the row, then the educational level options should be indented slightly so the reader can easily see the nested options.

Figure 2 shows that following adjustment (for father education and offspring sex) only fathers smoking after age 15 was associated with increased offspring BMI, but starting smoking before the age of 15 or after birth of offspring was not statistically associated. The strength of the association is modest and is demonstrated in Table 2. The use of both a table and figure for this result is perhaps unnecessary.

Table 3 and Fig 3 show the results for Father smoking and offspring FMI. Figure 4 provides interesting sex differences according to fathers smoking onset. There are very wide confidences on most of these data.

Maternal smoking onset and offspring BMI and FMI is presented in the same manner as paternal. Aternal smoking appears to be worse for the offspring in regards to the outcomes presented here.

The mediation analyses of father and mother smoking onset and offspring BMI is of interest and is well presented.

FigS2, along with all mediation plots, should be improved to capitalise the first letters of the Y-axis labels. Again, confidences are large in these data.

The authors should define what exactly the natural direct and indirect effects mean, so that those not used to these analysis can understand better and learn.

Discussion and conclusion:

The descriptive results are interesting, and may well be novel, however we must ask if they of fundamental importance. We know that parents smoking and being of high BMI is a huge risk factor for their children subsequently smoking and being of high BMI, this is well established. Children being of high BMI is a huge risk factor for being of high BMI in adulthood. So demonstrating that a fathers smoking influenced the BMI of their offspring in adulthood does seem somewhat intuitive. Despite this, it is of course important to prove such associations as the authors have done here.

See papers here that may remove some novelty from the findings in this manuscript. There appear to be others also and these should be discussed in the text, even if briefly.

http://dx.doi.org/10.1136/bmjopen-2015-007682

https://dx.doi.org/10.1038%2Fijo.2013.101

The discussion re possible biological mechanisms, focused on epigenetics, of the effects is sound.

The discussion of limitations is good, and highlights many of the issues. The major issue with this paper is the limited adjustment for confounding mediated through variables not considered in the analysis. The mediation of association analysis is interesting and of reportable note, but the primary findings remain largely unadjusted for confounders that are usually considered in the primary analysis. Some measures of risk factors usually included in models as utilised here include variables for physical (in)activity, nutritional status, alcohol consumption, sleep, stress, environmental factors such urban vs regional vs rural etc. A great deal of power is placed on the single measure of socio-economic status (parental education and offspring education) which is not ideal.

Despite this, the manuscript reports interesting findings in relation to the effect of parents pack-years on offspring BMI and FMI, however the headline findings that - fathers beginning to smoke before conception was associated with higher BMI in their adult sons and daughters, and that father’s smoking starting in any time window was associated with higher FMI in adult sons. And that mothers’ pre-as well as postnatal smoking onset was associated with higher offspring adult BMI, but not higher fat mass – are results relying on largely unadjusted models which suggest profound epigenetic influence across the life course. This must be presented and discussed with caution.

The authors appropriately mention the multi-factorial aspects of risk contributing to obesity in the conclusion and I think this is of vital importance.

Reviewer #2: Results

Lines 217-219, check the percentages and they were not adding up to 100%

Lines 250-251. It is not clear if non-significant associations for postnatal smoking or smoking onset <15 were due to reduced sample size in Table 2. They were both positive. Suggest adding N in each line of Table 2.

6. PLOS authors have the option to publish the peer review history of their article (what does this mean?). If published, this will include your full peer review and any attached files.

Reviewer #1: Yes: Henry West

Reviewer #2: No

---

## [Author Response · Author response to Decision Letter 0]

29 May 2020

Response to Reviewers 

First, we would like to thank the academic editor and reviewers for their valuable and interesting comments to improve the paper. Please find below our point-by-point response to the comments. 

The changes from the original version are referred to by page- and line numbers in the revised version of the manuscript (without track changes). 

In-house editor after resubmitting revised manuscript

Comment 1: Your manuscript files have been checked in-house but before we can proceed we need you to address the following issues: Thank you for including your ethics statement on the online submission form. To help ensure that the wording of your manuscript is suitable for publication, would you please also add this statement at the beginning of the Methods section of your manuscript file.

Response 1: Thank you for help ensuring that the wording in the manuscript is suitable for publication. In the revised version of the manuscript, we have added this statement in the first paragraph of the method section (lines 123-125) , and ethic committee name and approval number for each study center have been provided as supportive information in table S1 Resource ethics (line 125).

Comment 2: We also note the following comments in your cover letter: “Due to Norwegian ethical and legal restrictions the data underlying this study are available upon request to qualified researchers. Requests for data access can be directed to Haukeland University Hospital, 5021 Bergen, Norway. Att. RHINESSA PI Cecilie Svanes; email: postmottak@helse-bergen.no; phone: +47 55975000. Org, nr. 983 974 724.”

However, we see Dr. Cecilie Svanes is listed as an author of this study. To ensure your submission adheres to the PLOS ONE policy on acceptable data access restrictions, please provide non-author contact information for a data access committee, ethics committee, or other institutional body to which data requests may be sent. Note that it is not acceptable for an author to be the sole named individual responsible for ensuring data access.

Response 2: Thank you for drawing our attention to this inconsistency. In the revised cover letter, non-author contact information for data access has been provided.

Academic editor

Comment 1: Please ensure that your manuscript meets PLOS ONE's style requirements, including those for file naming.

Response 1: Thank you for drawing our attention to any inconsistency regarding style requirements. We have urged to adhere to PLOS ONE’s requirements and templates in the revised version of the manuscript.

Comment 2: Please include additional information regarding the survey or questionnaire used in the study and ensure that you have provided sufficient details that others could replicate the analyses. For instance, if you developed a questionnaire as part of this study and it is not under a copyright more restrictive than CC-BY, please include a copy, in both the original language and English, as Supporting Information.

Response 2: Thank you for bringing this to our attention. The questionnaires used in the ECRHS, RHINE and RHINESSA studies are not restricted by copyright more restrictive than CC-BY, and all questionnaire forms can be found and downloaded at the study cohorts web sites. This information has now been added in the method section in the revised manuscript, with URL provided so that all readers can easily access the questionnaires. 

Comment 3: We note that you have indicated that data from this study are available upon request. PLOS only allows data to be available upon request if there are legal or ethical restrictions on sharing data publicly.

Response 3: Due to Norwegian ethical and legal restrictions the data underlying this study are available upon request to qualified researchers. Requests for data access can be directed to Haukeland University Hospital, 5021 Bergen, Norway. Att. Head of Department. Dept. of Occupational Medicine, Marit Grønning; email: postmottak@helse-bergen.no; phone: +47 55975000. Org, nr. 983 974 724. The updated Data Availability statement is now promptly addressed in the revised cover letter.

Comment 4: We note you have included a table to which you do not refer in the text of your manuscript. Please ensure that you refer to Table 1 in your text; if accepted, production will need this reference to link the reader to the Table.

Response 4: Thank you for drawing our attention to this. In the revised version of the manuscript, we have removed the table that was not referred to in the text (regarding Resource. Funding). The funding information has been entered in the online submission system. 

Reviewer #1

Comment 1: The major revision I suggest is the acknowledgement of the myriad cardiovascular and endocrine risk factors that are all confounders to the outcomes reported here, and as such must be acknowledged as not adjusted for in this work.

Response 1: The reviewer is absolutely correct that a myriad of cardiovascular and endocrine risk factors could be influencing offspring BMI. However, our exposure of interest (parental smoking onset) is very far back in time – as far back as in the youth of the preceding generation. Since a confounder must per definition precede both exposure and outcome in time, this rules out many of the likely factors we would traditionally include in models assessing associations with BMI in adults. Any cardiovascular and endocrine risk confounders would have to be either in the parents’ childhood or as far back as in the grandparents to merit status as confounders. As far as we know, no studies have shown associations between children’s cardiovascular or endocrine characteristics and BMI in the next generation of adults – or between grandparental cardiovascular and endocrine characteristics and BMI in adult grandchildren. Some factors could perhaps have been included as potential mediators. However, since we already focus on four mediators (parental pack years, parental BMI, offspring smoking status and offspring birthweight), we feel it is difficult to include even more mediators without losing focus of the main aim, i.e. investigating relations between parental smoking onset in adolescence and BMI in the next generation. However, to properly acknowledge the myriad of cardiovascular and endocrine risk factors for adult BMI, we have added a paragraph in the revised Discussion where we discuss these and explain better why they are not confounders in the analysis of parental smoking onset in adolescence and adult offspring BMI (lines 446-449 and 451-452). 

Comment 2: The main point of this background is the suggestion that there is evidence that the germline cells of the parents might have critical exposure-sensitive periods for triggering epigenetic responses that will influence the risk of offspring becoming obese later in life. The remainder of the background is not directly of relevance to the manuscript but is interesting. It could be better placed in a broader discussion section as opposed to the background.

Response 2: We understand the reviewer’s comment. However, as one of the aims of the study was to investigate if effects were modified by offspring sex, we consider a paragraph addressing this in the background, to be of relevance.

Comment 3: Figure 1 is necessary and useful, but can be improved with more details provided about study centres and dates of the original surveys. Currently all we learn from the figure is sample numbers, when it is easy to make this figure more appealing to include the locations of these participants and the dates of the original studies and the form of the studies (postal questionnaires, clinics etc). 

Response 3: Thank you for pointing this out. We agree with the reviewer that additional details on study centres and survey dates could be beneficial. Information about time points, locations and form of the studies has been included in a new flow chart in the revised manuscript. 

Comment 4: Parents who started smoking in the 2 year interval around pregnancy were excluded, as the exposure seems only to be defined by years, and not by dates exactly. This is likely due to the original surveys only allowing answers by year.

Response 4: The reviewer is correct, and this is why we have not been able to define the exposure in more detail.

Comment 5: BMI from self-report. Less than ideal. Body fat mass from bioelectrical impedance analysis and FMI from that measurement/height^2. I am unsure how accurate such techniques are and as such, how valid this approach is. 

Response 5: We do agree with the reviewer that BMI based on self-reported height and weight is not ideal, and that is why we pointed this out as a limitation of the present study (lines 425-426). However, bioelectrical impedance analysis (BIA) has since it was introduced in the 1980’s been widely applied and particularly useful in large epidemiological studies, where more advanced methods are not feasible, and simpler methods are needed. In addition to the cohorts used in the present study (ECRHS, RHINE, and RHINESSA), BIA has also been used in other large cohort studies (NHANES (USA), NUGENOB (EU), MONICA (DK) to predict body composition (as fat mass and body fat %). Several studies have been conducted on the validation of BIA, and there is broad consensus that is a valid and precise tool for estimating body composition in healthy subjects. We therefore consider this method useful and a valid measure of obesity and to compare body composition across populations (https://www.nature.com/articles/ejcn2012168;
https://academic.oup.com/ajcn/article/64/3/459S/4651645;
https://academic.oup.com/ajcn/article/64/3/436S/4651636) 

Comment 6: The variables considered were parental socioeconomic status (via parental education), parental pack years, parental BMI (self report), offspring smoking (ever vs never only) and offspring birthweight. This is a small number of confounders. These are all indeed vital confounders to consider, but there are myriad other critical confounders that could be influencing the outcomes from across the huge list of cardiovascular and endocrine risk factors both prior to conception and post-natally.

Response 6: We acknowledge this concern. However, as outlined in our response to comment 1 above, a confounder must precede both exposure and outcome in time. For a more elaborate response, please see our response to comment 1. 

Comment 7: We also know that the amount of smoking in close confinement to the offspring post birth is of importance, as studies with serum cotinine samples of tobacco exposure prove that although some parents may be smoking a great deal (as per pack years), if they do so outside or otherwise away from the child, they can mitigate possible exposures to their offspring.

Response 7: We thank the reviewer for pointing out this important issue. Unfortunately, we do not have any information about where the parents smoked so we cannot shed proper light on this in our analyses. Nevertheless, since this is of high relevance for our study, we have added a paragraph in the revised discussion with some reflections regarding this limitation (lines 438-440). 

Comment 8: In reality a number of other confounders not listed in the methods were also included in results: parental age, offspring education, other parents smoking habits, and other parents BMI (other parents information obtained from the offspring…), however these were not include din final models.

Response 8: The reviewer correctly states that we did consider these factors as potential confounders, and we referred to these in the DAG (figure S1). However, we did not include them in the final models since they could not be defined as true confounders, i.e. preceding both exposure and outcome in time. For more information regarding this, please see our response to comment 1 above. 

Comment 9: Table 1A and 1B and S1A and B: significant missing values for age smoking onset and packyears noted.

Response 9: The reviewer correctly states that there is a significant amount of missing values for packyears in the tables. However, the number of missing values for age of smoking is in fact not correctly updated, and we thank the reviewer for drawing our attention to this. In the final study population, where subjects with missing information on parental smoking have been excluded, these numbers equal 0. This has been corrected in the revised manuscript.

Comment 10: Tables would be clearer with stepped rows under umbrella terms. For example: ‘educational level, n (%)’ should be far left of the row, then the educational level options should be indented slightly so the reader can easily see the nested options. 

Response 10: We agree with the reviewer, and in the revised manuscript, we have changed table 1A and 1B according to the comment.

Comment 11: Figure 2 shows that following adjustment (for father education and offspring sex) only fathers smoking after age 15 was associated with increased offspring BMI, but starting smoking before the age of 15 or after birth of offspring was not statistically associated. The strength of the association is modest and is demonstrated in Table 2. The use of both a table and figure for this result is perhaps unnecessary.

Response 11: We acknowledge the reviewer’s comment that it may be unnecessary to present both a table and a figure of the adjusted association between parental smoking onset and offspring bmi. However, as the figure additionally visualise both the crude and adjusted regression analyses (the table presents adjusted analysis), we do think including both of them adds information on the strength and the direction of the effect estimates, which may be of interest to a reader.

Comment 12: FigS2, along with all mediation plots, should be improved to capitalise the first letters of the Y-axis labels.

Response 12: Thank you for pointing this out. In the revised manuscript, all the mediation plots have capital first letter of the Y-axis labels.

Comment 13: The authors should define what exactly the natural direct and indirect effects mean, so that those not used to these analysis can understand better and learn.

Response 13: We agree with the reviewer that the methodology could be explained in a clearer manner. In the revised paragraph, we have attempted to improve the definition of natural direct and indirect effects so that readers may more easily understand (lines 211-213). 

Comment 14: The descriptive results are interesting, and may well be novel, however we must ask if they of fundamental importance. We know that parents smoking and being of high BMI is a huge risk factor for their children subsequently smoking and being of high BMI, this is well established. Children being of high BMI is a huge risk factor for being of high BMI in adulthood. So demonstrating that a fathers smoking influenced the BMI of their offspring in adulthood does seem somewhat intuitive. Despite this, it is of course important to prove such associations as the authors have done here. See papers here that may remove some novelty from the findings in this manuscript. There appear to be others also and these should be discussed in the text, even if briefly.

Response 14: We appreciate that the results may seem somewhat intuitive, as the reviewer points out. Nevertheless, we believe that the results are highly novel because we look at smoking in one generation and BMI in the next generation, suggesting epigenetic mechanisms instead of traditional exposure-outcome associations within the individual. We agree that such associations are known for BMI (that parental BMI is associated with offspring BMI), and that smoke exposure in an individual (active smoking, but also passive) is associated with BMI in the same individual. However, evidence is very scarce with regard to inter-generational effects of smoking on BMI. We thank the reviewer for the two relevant papers suggested. One of them is included in the background section (line 89), and the other is added in the discussion section (line 410).

Comment 15: The major issue with this paper is the limited adjustment for confounding mediated through variables not considered in the analysis. The mediation of association analysis is interesting and of reportable note, but the primary findings remain largely unadjusted for confounders that are usually considered in the primary analysis. Some measures of risk factors usually included in models as utilised here include variables for physical (in)activity, nutritional status, alcohol consumption, sleep, stress, environmental factors such urban vs regional vs rural etc. A great deal of power is placed on the single measure of socio-economic status (parental education and offspring education) which is not ideal.

Response 15: We acknowledge this concern. For a more elaborate response, please see our response to comment 1. 

Comment 16: Despite this, the manuscript reports interesting findings in relation to the effect of parents pack-years on offspring BMI and FMI, however the headline findings that - fathers beginning to smoke before conception was associated with higher BMI in their adult sons and daughters, and that father’s smoking starting in any time window was associated with higher FMI in adult sons. And that mothers’ pre-as well as postnatal smoking onset was associated with higher offspring adult BMI, but not higher fat mass – are results relying on largely unadjusted models which suggest profound epigenetic influence across the life course. This must be presented and discussed with caution. 

Response 16: We acknowledge this concern, and therefore point out in the discussion that “A multitude of exposures and difference in genetic background exists in population studies, and as the offspring in the present study have reached adulthood, they have been exposed to a variety of environmental factors”(lines 443-446)”Thus, unmeasured factors may have impacted on our findings” (line 454). With regard to the mediation analysis, we also acknowledge that we have not been able to investigate “whether there are violations to the identification assumptions, especially with regard to all potential variables being independent and accounted for” (lines 457-458). This therefore needs further investigation. With regard to the reviewer’s comment on largely unadjusted models, for a more elaborate response, please see our response to comment 1. 

Reviewer #2

Comment 1: Lines 217-219, check the percentages and they were not adding up to 100%.

Response 1: Thank you for bringing to our attention that this should be better described. The percentages are calculated based on number of unique parents (fathers=2111, mothers =2569) and not on number of participating offspring. In the revised version of the manuscript this has been pointed out in the text (line 223) and in table 1A (line 238) and table 1B (line 242) to make it easier for the readers to understand what the numbers are based on. 

Comment 2: Lines 250-251. It is not clear if non-significant associations for postnatal smoking or smoking onset <15 were due to reduced sample size in Table 2. They were both positive. Suggest adding N in each line of Table 2.

Response 2: We agree with the reviewer, and this has been included in the table in the revised manuscript

---

## [Editor Report · Decision Letter 1]

19 Jun 2020

Parents’ smoking onset before conception as related to body mass index and fat mass in adult offspring: Findings from the RHINESSA generation study

PONE-D-20-07665R1

Dear Dr. Knudsen,

We’re pleased to inform you that your manuscript has been judged scientifically suitable for publication and will be formally accepted for publication once it meets all outstanding technical requirements.

Kind regards,

Seana Gall

Academic Editor

PLOS ONE

Additional Editor Comments (optional):

Thank you for your detailed responses to the editor and reviewer comments. They were well considered.
---

## [Editor Report · Acceptance letter]

24 Jun 2020

PONE-D-20-07665R1 

Parents’ smoking onset before conception as related to body mass index and fat mass in adult offspring: Findings from the RHINESSA generation study 

Dear Dr. Knudsen:

I'm pleased to inform you that your manuscript has been deemed suitable for publication in PLOS ONE. Congratulations! Your manuscript is now with our production department. 

Kind regards, 

on behalf of

Dr. Seana Gall 

Academic Editor

PLOS ONE